# Optimization Strategies Aimed to Increase the Efficacy of *Helicobacter pylori* Eradication Therapies with Quinolones

**DOI:** 10.3390/molecules25215084

**Published:** 2020-11-02

**Authors:** Javier P. Gisbert

**Affiliations:** Gastroenterology Unit, Hospital Universitario de La Princesa, Instituto de Investigación Sanitaria Princesa (IIS-IP), Universidad Autónoma de Madrid (UAM), and Centro de Investigación Biomédica en Red de Enfermedades Hepáticas y Digestivas (CIBEREHD), 28006 Madrid, Spain; javier.p.gisbert@gmail.com; Tel.: +34-913093911; Fax: +34-915204013

**Keywords:** *Helicobacter pylori*, optimized, optimization, quinolones, levofloxacin, moxifloxacin, sitafloxacin

## Abstract

*H. pylori* infection is the main cause of gastritis, gastroduodenal ulcer disease, and gastric cancer. Fluoroquinolones such as levofloxacin, or more recently moxifloxacin or sitafloxacin, are efficacious alternatives to standard antibiotics for *H. pylori* eradication. The aim of the present review is to summarize the role of quinolone-based eradication therapies, mainly focusing on the optimization strategies aimed to increase their efficacy. Several meta-analyses have shown that, after failure of a first-line eradication treatment, a levofloxacin-containing rescue regimen is at least equally effective, and better tolerated, than the generally recommended bismuth quadruple regimen. Compliance with the levofloxacin regimens is excellent, and the safety profile is favourable. Higher cure rates have been reported with longer treatments (>10–14 days), and 500 mg levofloxacin daily is the recommended dose. Adding bismuth to the standard triple regimen (PPI-amoxicillin-levofloxacin) has been associated with encouraging results. Unfortunately, resistance to quinolones is easily acquired and is increasing in most countries, being associated with a decrease in the eradication rate of *H. pylori*. In summary, a quinolone (mainly levofloxacin)-containing regimen is an encouraging second-line (or even third-line) strategy, and a safe and simple alternative to bismuth quadruple therapy in patients whose previous *H. pylori* eradication therapy has failed.

## 1. Introduction

*Helicobacter pylori* (*H. pylori*) is a worldwide infection that is the main cause of gastric cancer and gastroduodenal ulcer disease [1]. Recent clinical trials and meta-analyses have evidenced that the most commonly used first-line therapies—a proton pump inhibitor (PPI) plus two antibiotics generally including clarithromycin and either amoxicillin or metronidazole—fail in more than 20–30% of patients [2]. One of the major factors affecting our ability to cure *H. pylori* infection is antibiotic resistance, mainly to clarithromycin, which seems to be increasing in many geographic areas [3].

A rescue regimen including a quadruple combination of a PPI, bismuth, tetracycline, and metronidazole has been used as the optimal second-line approach after initial *H. pylori* eradication failure [4,5,6]. However, this regimen fails to eradicate the infection in at least 20% of cases [7,8,9,10]. In addition, administration of this regimen is relatively complex, is associated with a high incidence of adverse events, and many countries are currently experiencing a general unavailability of tetracycline and/or bismuth.

On the other hand, the fluoroquinolones have a broad spectrum of activity against Gram-negative and Gram-positive bacteria and atypical respiratory pathogens [11]. In particular, several studies have demonstrated the efficacy of quinolones in the treatment of infections of the respiratory tract, genitourinary tract and skin [11]. Recent findings indicate that some fluoroquinolones such as levofloxacin, or more recently moxifloxacin or sitafloxacin, seem to be efficacious alternatives to standard antibiotics, mainly as rescue regimens after initial *H. pylori* eradication failure.

Thus, at present, quinolone-containing rescue regimens represent an encouraging strategy for *H. pylori* rescue treatment, as some studies have demonstrated that, for example, levofloxacin has remarkable in vitro activity against this bacteria [12]. Moreover, levofloxacin retains its activity, in vitro, when *H. pylori* strains are resistant to clarithromycin and metronidazole [13,14]. These encouraging results have been confirmed in vivo, showing that most patients with both metronidazole and clarithromycin *H. pylori* resistance, are cured with the levofloxacin-containing regimen [15,16,17].

The aim of the present review is to summarize the role of quinolones in the management of *H. pylori* infection, mainly focusing on the optimization strategies aimed to increase the efficacy of quinolone-based eradication therapies.

## 2. Bibliographic Search

A systematic bibliographic search was designed to identify studies evaluating the role of quinolones in the management of *H. pylori* infection, mainly those investigating optimization strategies aimed to increase the efficacy of eradication therapies. An electronic search was performed in PubMed up to October 2020 using the following algorithm: (quinolone[TI] OR levofloxacin[TI] OR moxifloxacin[TI] OR sitafloxacin[TI]) AND (*Helicobacter pylori* OR *H. pylori*). In addition, the reference lists from the selected articles were reviewed to identify additional studies of potential interest. Articles published in any language were included.

## 3. Meta-Analyses Evaluating the Efficacy of Quinolone-Based Regimens for *H. pylori* Eradication

Several meta-analyses have evaluated the efficacy of quinolone-based *H. pylori* eradication regimens, their main characteristics and conclusions being summarized in Table 1 [18,19,20,21,22,23,24,25,26,27,28,29,30,31,32,33,34]. Initially, in 2006, two meta-analyses suggested that, after failure of treatment to eradicate *H. pylori*, a levofloxacin-containing rescue regimen was at least equally (and perhaps even more) effective, and better tolerated, than the generally recommended bismuth-containing quadruple regimen (i.e., a PPI, bismuth, tetracycline, and metronidazole) [18,19]. Further studies evaluating mainly a levofloxacin-containing triple rescue combination, revealed a mean eradication rate of approximately 80%. Therefore, at present, a combination of a PPI, levofloxacin and amoxicillin (perhaps with the addition of bismuth) constitutes an encouraging alternative for *H. pylori* rescue treatment.

## 4. Advantages of Levofloxacin-Based Regimens for *H. pylori* Eradication

As previously mentioned, administration of the classic bismuth-containing quadruple regimen is complex (tetracycline is usually prescribed every six hours and metronidazole every 8 h). On the other hand, levofloxacin-based regimens (with the PPI and amoxicillin administered twice daily, and levofloxacin every 24 h) represent an encouraging simpler alternative. In our previous experience, compliance with this regimen is excellent, with more than 95% of the patients taking all the medications correctly [35,36].

Another advantage is that quinolones in general, and levofloxacin in particular, are generally well tolerated, and most associated adverse events are mild and transient [11]. In this respect, adverse effects have been reported in approximately 20% of our patients, the most common being gastrointestinal effects [35,36]. Only in approximately 2% of the cases the adverse effects were classified as severe, but symptoms were limited to the duration of treatment in most patients. Thus, in a recent systematic review of the literature, the incidence of adverse effects with levofloxacin-based regimens was 18%, and only 3% were severe [18], which is consistent with our results. Finally, it should be stressed that, as previously mentioned, several meta-analyses have confirmed a lower incidence of adverse effects with levofloxacin-containing regimens compared with the classic bismuth quadruple combinations [18,19,22].

In 2018, the United States Food and Drug Administration (FDA) and the European Medicines Agency (EMA) issued new warnings about serious adverse effects of fluoroquinolones [37]. Both the FDA and the EMA demanded a change in the labelling of the entire class of antibiotics highlighting these new risks. They also discouraged the use of fluoroquinolones for most mild and moderate infections or where there is a therapeutic alternative, restricting their use exclusively to serious infections such as pneumonia, anthrax and plague, or other non-self-limiting infections in which the therapeutic benefit outweighs the risks [37]. Nevertheless, it is important to remember that *H. pylori* causes a chronic infection that can trigger serious conditions such as peptic ulcer (possibly with complications, such as gastrointestinal bleeding) and gastric cancer, and that treatment requires the combined use of several antibiotics. In any case, this health authority recommendation supports the responsible use of antibiotics and the need to carefully analyze the available therapeutic options and their risk/benefit ratio before prescribing any drug.

In summary, a levofloxacin-containing regimen is an encouraging second-line (or even third-line) strategy, and a safe and simple alternative to bismuth quadruple therapy in patients whose previous *H. pylori* eradication therapy has failed. Accordingly, several international guidelines (Maastricht V, Toronto and American College of Gastroenterology Consensus Reports) have recommended this quinolone-based regimen as a second-line rescue option after failure of some first-line regimens, such as standard triple, concomitant or bismuth quadruple [7,8,9].

## 5. *H. pylori* Resistance to Quinolones

The prevalence of antibiotic resistance to *H. pylori* is the main factor affecting the efficacy of current therapeutic regimens, and it is increasing worldwide [3]. Unfortunately, resistance to quinolones in general, and to levofloxacin in particular, is easily acquired; as a consequence, the resistance rate is increasing and already relatively high in countries with a high consumption of these drugs [38,39,40]. In a recent systematic review on the prevalence of resistance of *H. pylori* to antibiotics in different countries, the overall calculated levofloxacin resistance rate was 16%, although the figures varied significantly depending on the continent [41]. In particular, the prevalence rate was higher in Europe (24%) as compared to Asia (11.6%), and it was absent in Africa [41]. In Asia, different values among countries were detected, the resistance rate being 14.9% in Japan, 11.9% in Taiwan, and 2.6% in Hong Kong [41]. The prevalence of primary antibiotic resistance of *H. pylori* in 2008 and 2009 in 18 European countries was assessed in a multicenter study, and the rate for levofloxacin was 14% [3]. More recently, Savoldi et al. performed a systematic review and meta-analysis to assess the distribution of *H. pylori* resistance to commonly used antibiotics, and found that primary levofloxacin resistance in the European region was 11% [42]. In particular, primary levofloxacin resistance was ≥15% in all WHO regions, except in the European region (11%). Thus, this figure was 15% in the Americas region, 19% in the Eastern Mediterranean region, 30% in the Southeast Asia region, and 22% in the Western Pacific region [42]. This resistance rates are clinically relevant, as it has been shown that antibiotic resistance to quinolones causes a decrease in the eradication rate of *H pylori*. Thus, a recent systematic review and meta-analysis showed that the pooled relative risk of eradication rate in patients with *H. pylori* strains sensitive versus resistant to levofloxacin was 0.79 [43]. Accordingly, some studies have suggested that the efficacy of levofloxacin-containing therapy is decreasing, most likely due to increased primary resistance [44]. Furthermore, some consensus (such as Houston and Taipei consensus statements) state that fluoroquinolone, metronidazole, and clarithromycin-containing triple therapies should not be used empirically with the caveat that they could be considered if resistance was proven to be rare in a region [45,46].

The main mechanism of quinolone resistance is mutation of *gyrA* gene in *H. pylori.* Thus, *gyrA* mutation status has been demonstrated to be an important factor in predicting successful eradication with sitafloxacin-containing therapies [47,48]. In this respect, a recent meta-analysis concluded that the status of *gyrA* mutation was a superior marker for predicting successful eradication in sitafloxacin/amoxicillin-containing triple regimen as a third-line rescue therapy [34].

## 6. Pharmacokinetics of Quinolones

The 6-fluoroquinolones are synthetic antimicrobials that have good absorption from the gastrointestinal tract, and generally favorable pharmacokinetic properties. Thus, fluoroquinolones have many advantageous pharmacokinetic properties including high oral bioavailability, large volume of distribution, and broad-spectrum antimicrobial activity [49,50].

Ciprofloxacin, ofloxacin, levofloxacin, and moxifloxacin all have oral and intravenous formulations that allow direct estimates of oral bioavailability, with values of 70% for ciprofloxacin, 86% for moxifloxacin, and >95% for ofloxacin and levofloxacin [51]. In particular, the almost complete (≥99%) absolute oral bioavailability suggests that a comparable exposure to the intravenous regimen may be achieved after oral administration [51,52]. Norfloxacin has an oral formulation only, and its estimated bioavailability is approximately 30 to 40%. Quinolones (e.g., levofloxacin, sparfloxacin, gatifloxacin, and moxifloxacin) have elimination *t*_1/2_s that permit once-daily dosing and demonstrate pharmacokinetic profiles that are generally linear and dose proportional over the clinical-dose range [53,54,55]. Peak plasma concentration (C_max_) is usually attained 1–2 h after oral dosing. The plasma concentration profile after intravenous administration is comparable in area under the concentration-time curve (AUC) to that observed for oral tablets when equal doses are administered [56]. Finally, the pharmacokinetics of novel quinolones, such as garenoxacin, also supports once-daily administration [57].

Food does not substantially reduce fluoroquinolone absorption but may delay the time to reach peak serum concentrations [58,59]. However, dairy, antacids, multivitamins containing zinc, certain medications (e.g., sucralfate, buffered formulation of didanosine), and other sources of divalent cations (aluminum, magnesium, calcium) can substantially decrease absorption (presumably by formation of cation-quinolone complexes). Concurrent use should be avoided or these substances should be given several hours apart from the fluoroquinolone in order to avoid their interaction [60].

The volumes of distribution of quinolones are high and, in most cases, exceed the volume of total body water, indicating accumulation in some tissues. Concentrations in prostate tissue, stool, bile, lung, and neutrophils as well as macrophages usually exceed serum concentrations. Concentrations in urine and kidney tissue are high for the quinolones with a major renal route of elimination (all except moxifloxacin) [56]. Concentrations of quinolones in saliva, prostatic fluid, bone, and cerebrospinal fluid are usually lower than drug concentrations in serum.

## 7. How to Optimize Quinolone-Based Treatments for *H. pylori* Eradication?

The factors responsible for effective antimicrobial therapy of a *H. pylori* infection are both straightforward and easily discoverable [61,62,63]. Thus, the most common cause for treatment failure is the presence of organisms resistant to one or more of the antimicrobials prescribed (in addition to poor adherence to therapy). Ideally, infectious disease therapies in general, and *H. pylori* infection in particular, are chosen based on culture and susceptibility testing from each patient. However, culture is relatively expensive, not because of the cost of the procedure per se, but mainly because of the costs of the associated endoscopy required to obtain biopsy specimens [5]. Nevertheless, it is not needed to test every patient but rather it would only be necessary to know the local and regional rates of resistance (if high, then the individual susceptibility testing would be recommended). The alternative, on the other hand, is to empirically choose a regimen based on the local pattern of resistance [64].

Treatment-dependent variables are also important if we want to achieve a high eradication rate [61,63,65]. For example, for some antibiotics, resistance can be prevented, and at least partially overcome, by increasing the dose or the treatment duration. In this section we will review the optimization strategies aimed to increase the efficacy of quinolone-based eradication therapies.

### 7.1. Duration of Treatment

Duration of quinolone therapy, more than the dosage, seems to be the crucial factor affecting eradication rate [44]. Three meta-analyses [18,19,22] found, as did three recent randomized controlled trials [44,66,67], higher cure rates with 10 to 14-day than with 7-day levofloxacin-containing regimens. Furthermore, two recent studies have compared the efficacy of 14, 10 and 7-day levofloxacin-containing triple therapy as rescue regimen, and a higher eradication rate was demonstrated with the longest regimen [68,69].

On the other hand, recent studies have demonstrated that a 10-day triple therapy with moxifloxacin is more effective than the same treatment for only 5 or 7 days [70,71]. Finally, in a recent randomized controlled trial, 14 days of moxifloxacin treatment significantly increased the *H. pylori* cure rate compared with the 7-day regimen [72].

### 7.2. Antibiotic Dose and Frequency

For time-dependent antibiotics (e.g., amoxicillin), it is more important to prolong the time that the plasma concentration is higher than the minimal inhibitory concentration (MIC), rather than achieve higher drug levels. On the other hand, for concentration-dependent antibiotics (e.g., levofloxacin, clarithromycin and metronidazole), it is more important to achieve higher plasma levels [73].

Levofloxacin 500 mg daily has been demonstrated to be equally effective but better tolerated than higher doses (e.g., 1000 mg/day) [44,74,75,76]. Comparative studies of 500 mg, 750 mg, and 1 g of levofloxacin for 7 days or 10 days confirmed that duration of treatment was more important than quinolone dosage [44]. More recently, in a randomized controlled trial, levofloxacin 200 mg twice daily was found to be similar in efficacy for eradicating *H. pylori* infection to levofloxacin 500 mg once daily, but with lower mean total costs [77]. According to this, it has been observed that the dosage of levofloxacin cannot overcome levofloxacin resistance [78]. In summary, when prescribing a quinolone regimen, 500 mg levofloxacin daily should be enough to eradicate *H. pylori* infection.

### 7.3. Addition of Bismuth

Bismuth is one of the few antimicrobials to which resistance is not developed [79]. In addition, bismuth has an additive effect with antibiotics, overcomes levofloxacin and clarithromycin resistance and its efficacy is not affected by metronidazole resistance [79,80]. Therefore, combining bismuth and levofloxacin in the same regimen may be a promising option.

The mechanism of action of bismuth appears to be more antiseptic than antibiotic [81,82]. It has been suggested that bismuth exerts its antibacterial action mainly by preventing bacterial colonization and adherence to gastric epithelium and by binding toxins produced by *H. pylori* [83]. In addition, bismuth decreases mucin viscosity, reduces the bacterial load and has a synergistic effect with antibiotics [79]. In this respect, already in 1987 it was shown that the combination of bismuth subcitrate with the older quinolone, oxolinic acid, induced synergistic activity against *H. pylori* [84].

Some authors have evaluated a combination of a triple therapy with a PPI-amoxicillin-levofloxacin but adding bismuth and thus converting this triple regimen into a quadruple one, with encouraging results (Table 2) [77,80,85,86,87,88,89,90,91,92,93], generally better than those obtained by previously published studies with levofloxacin triple therapies [18,19,22,31]. One of these levofloxacin-bismuth studies was focused specifically in patients with one previous *H. pylori* eradication failure (the most common scenario for the use of quinolones in clinical practice), achieving an eradication rate of 90% [89] which may be considered encouraging, especially taking into account that this rescue regimen was prescribed empirically [89]. Only two studies reported suboptimal cure rates of only approximately 70% with levofloxacin-bismuth quadruple therapy, which might be explained by inclusion of patients with one or more failures of eradication therapies, of whom some had previously received levofloxacin [92,93].

With respect to the additive/synergistic effect of bismuth, two randomized controlled trials have shown that the addition of this compound to a triple therapy that included a PPI, amoxicillin and levofloxacin or moxifloxacin, increased the *H. pylori* eradication rate [80,94]. In the study by Liao et al., patients were randomized to receive a PPI, amoxicillin and levofloxacin with or without bismuth for 14 days, and it was found that the eradication rate was slightly higher with the bismuth-containing regimen (87% vs. 83%); however, the most remarkable finding was that the bismuth combination was still relatively effective (71%) for levofloxacin-resistant strains, while the non-bismuth regimen achieved *H. pylori* eradication in only 37% of the cases [80]. Nevertheless, whether the outcome of adding bismuth is really additive or merely synergistic is still debated, and therefore the role of adding bismuth to the conventional 14-day levofloxacin-based triple regimen remains unclear.

Non-bismuth quadruple, either sequential and concomitant, regimens including a PPI, amoxicillin, clarithromycin and a nitroimidazole, are increasingly used as first-line treatments as they are considerably effective [95]. However, following failure of these regimens, the best empirical rescue therapy remains unknown. These patients have limited options for further therapy because they already have received three different relevant antibiotics such as clarithromycin, amoxicillin and metronidazole. In a recent study, cure rates with the levofloxacin-bismuth quadruple rescue regimen were similar when compared depending on the previous treatment (standard triple therapy 88% vs. sequential 94% vs. concomitant 92%) [89]. Therefore, the levofloxacin-bismuth-containing quadruple therapy constitutes an encouraging second-line strategy in patients with non-bismuth quadruple sequential or concomitant treatment failure, achieving better results than those previously obtained with levofloxacin triple therapy [96,97,98,99,100]. Thus, the eradication rate of a 10-day PPI-amoxicillin-levofloxacin therapy after the failure of concomitant and sequential treatment was of approximately 80% in a recent meta-analysis [101], while optimization of the regimen through addition of bismuth increased cure rates by 10%, reaching the generally recommended 90% threshold [89].

Adverse events associated with the bismuth-levofloxacin-amoxicillin treatment have been relatively frequent, but in only a very low proportion of the cases were these adverse events classified as intense, and none of them was classified as serious adverse event. Accordingly, treatment withdrawal due to levofloxacin related adverse events has been exceptional [18,80,85,88,89,90,93]. Regarding bismuth safety, the doses currently used for *H. pylori* eradication in the quadruple regimen are relatively low and are prescribed for a short time period, leading to safe blood levels [102]. Accordingly, when comparing a levofloxacin-containing triple therapy with or without the addition of bismuth, no significant difference in the incidence of side effects was shown [80].

In summary, 14-day bismuth plus levofloxacin-containing quadruple therapy may be considered a safe and effective rescue strategy for patients whose previous standard triple or non-bismuth quadruple (either sequential or concomitant) therapies have failed, providing a more effective option than classic bismuth-quadruple or levofloxacin-triple standard regimens.

## 8. The Experience with Quinolones in the “European Registry on *H. pylori* Management”

The “European Registry on *H. pylori* Management” (Hp-EuReg) is an international multicenter prospective non-interventional registry starting in 2013 aimed to evaluate the decisions and outcomes in *H. pylori* management by European gastroenterologists [103]. Thirty European countries, with over 300 recruiters, are actively participating in this project, where patients are managed and registered according to their routine clinical practice. Very recently, the patterns and trends in first-line empirical eradication prescription and outcomes of five years and 21,533 patients have been published [10].

In the Hp-EuReg, more than 400 received a levofloxacin triple therapy as a first-line regimen, and the eradication rate was of only 50% approximately [10], which may be due, at least in part, to the relatively high quinolone resistant rates in Europe [41] (in the Hp-EuReg, approximately 20% of the naïve patients had already levofloxacin resistance) [104]. On the other hand, to evaluate the effectiveness of second-line empirical treatments in the Hp-EuReg, overall, 4862 patients were studied [105]. After failure of first-line clarithromycin-containing treatment, optimal eradication (>90%) was obtained with moxifloxacin-containing triple therapy, three-in-one single capsule (Pylera^®^) or quadruple therapy with levofloxacin and bismuth. With this last quadruple regimen, which was prescribed to more than 500 patients, 90% *H. pylori* eradication rate was achieved. In patients receiving triple regimens containing levofloxacin or the standard bismuth quadruple regimen, cure rates were optimized with 14-day regimens using high doses of PPIs. Therefore, from the results of the Hp-EuReg it can be concluded that empirical second-line triple therapies generally provided low eradication rates except when prescribing 14 days of levofloxacin or moxifloxacin. However, high effectiveness was obtained with second-line bismuth-containing quadruple therapies [105].

One step forward, to evaluate the effectiveness of empirical rescue therapies on third and subsequent lines in Europe, 1782 rescue treatments were included: 1264, 359, 125 and 34 third-, fourth-, fifth- and sixth-line treatments, respectively [106]. Three regimens achieved an optimal eradication rate (≥90%): three-in-one single capsule (Pylera^®^), quadruple PPI-bismuth-tetracycline-metronidazole and triple PPI-amoxicillin-levofloxacin, the two latter only when high PPI doses and 14 days’ treatment duration were used.

Penicillin allergy is the most common type of drug allergy, reported in about 5–10% of individuals. However, to date, only few studies have evaluated the efficacy of first-line *H. pylori* eradication treatment specifically in patients allergic to penicillin. Furthermore, the appropriate rescue therapy when eradication therapy fails in this scenario has not been properly evaluated. Thus, more than 1000 patients allergic to penicillin from the Hp-EuReg were analyzed [107]. In second-line, after the failure of a clarithromycin-metronidazole-containing regimen, a combination of a PPI-clarithromycin and levofloxacin achieved *H. pylori* eradication in 71% of the cases [107].

Finally, regarding the safety profile of quinolone therapy in the Hp-EuReg, adverse events related to levofloxacin were relatively frequent (approximately 20%, most of them related to the gastrointestinal system, including nausea and diarrhoea) but generally mild, with a very low (0.1%) percentage of serious adverse events [108]. These results are fully coincident with those reported by the largest series of levofloxacin-based treatments [100] and with a previous systematic review [18]. Furthermore, as previously mentioned, several meta-analyses have confirmed a lower incidence of adverse events with levofloxacin-based regimens than with the classic bismuth quadruple combination. Finally, in a network meta-analysis comparing tolerance of treatments for *H. pylori*, all regimens were considered tolerable, but 7 days of levofloxacin-based triple treatment ranked best in terms of the proportion of adverse events reported [30]. Nevertheless, based on the experience of the Hp-EuReg, the risk of suffering from adverse events increased with longer durations of the treatment, from 21% for 7 days to 39% for 14 days [108].

## 9. Conclusions

Fluoroquinolones such as levofloxacin, or more recently moxifloxacin or sitafloxacin, seems to be efficacious alternatives to standard antibiotics (such as clarithromycin or metronidazole), mainly as rescue regimens after initial *H. pylori* eradication failure. Several meta-analyses have evaluated the efficacy of quinolone-based *H. pylori* eradication regimens and have concluded that, after failure of treatment to eradicate *H. pylori*, a levofloxacin-containing rescue regimen is at least equally effective, and better tolerated, than the generally recommended bismuth quadruple regimen. In fact, compliance with the levofloxacin regimen is excellent, and this treatment is generally well tolerated (most associated adverse events are mild and transient). Furthermore, levofloxacin-based regimen represents a simpler alternative compared with bismuth quadruple therapy. Higher cure rates with 10 and mainly 14-day levofloxacin-containing regimens, compared with 7-day treatments, have been reported. When prescribing a quinolone regimen, 500 mg levofloxacin daily should be enough to eradicate *H. pylori* infection. Adding bismuth and thus converting the standard triple regimen (PPI-amoxicillin-levofloxacin) into a quadruple one, has been associated with encouraging results. Unfortunately, resistance to quinolones in general, and to levofloxacin in particular, is easily acquired, and in countries with a high consumption of these drugs, the resistance rate is increasing and is already relatively high, being associated with a decrease in the eradication rate of *H. pylori*. In summary, a quinolone (mainly levofloxacin)-containing regimen is an encouraging second-line (or even third-line) strategy, and a safe and simple alternative to bismuth quadruple therapy in patients whose previous *H. pylori* eradication therapy has failed.

## Figures and Tables

**Table 1 molecules-25-05084-t001:** Meta-analyses evaluating the efficacy of *H. pylori* quinolone-based regimens.

Author	Year	Number of Studies Included	First-Line or Rescue Regimen	Quinolone Type	Conclusions
Gisbert [18]	2006	14	Rescue	Levofloxacin	After *H. pylori* eradication failure, levofloxacin-based rescue regimen is more effective and better tolerated than the generally recommended quadruple therapy. A 10-day combination of PPI-levofloxacin-amoxicillin constitutes an encouraging second-line alternative
Saad [19]	2006	4	Rescue	Levofloxacin	A 10-day course levofloxacin triple therapy is more effective and better tolerated than 7-day bismuth-based quadruple therapy in the treatment of persistent *H. pylori* infection
Zhang [20]	2008	11	First-line	Levofloxacin	PPI and levofloxacin-based triple therapy is effective in the eradication of *H. pylori*, and should be advocated to be the first-line regime
Wenzhen [21]	2009	4	First-line	Moxifloxacin	Moxifloxacin-based triple therapy is more effective and does not increase the incidence of overall side effects compared to clarithromycin-based triple therapy in the treatment of *H. pylori* infection
Li [22]	2010	20	Rescue	Levofloxacin & moxifloxacin	Second-generation fluoroquinolone-based triple therapy can be suggested as the regimen of choice for rescue therapy in the eradication of persistent *H. pylori* infection especially 10-day levofloxacin-based triple therapy
Wu [23]	2011	7	Rescue	Moxifloxacin	Moxifloxacin-containing triple regimen is more effective and better tolerated than the bismuth-containing quadruple therapy in the second-line treatment of *H. pylori* infection
Di Caro [24]	2012	14	Rescue	Levofloxacin	Our findings support the use of 10-day levofloxacin-amoxicillin as a simple second-line treatment for *H. pylori* eradication with an excellent eradication rate and tolerability
Marin [25]	2013	19	Rescue	Levofloxacin	In a routine clinical practice setting, the most adequate second-line treatment consists in a 10-day regimen of PPI-levofloxacin-amoxicillin given twice daily, unless regional or new data show high quinolone resistance
Zhang [26]	2013	7	First-line & rescue	Moxifloxacin	Moxifloxacin-based triple therapy is more effective and better tolerated than standard triple or quadruple therapy. Therefore, a moxifloxacin-based triple regimen should be used in the second-line treatment of *H. pylori* infection
Peedikayil [27]	2014	7	First-line	Levofloxacin	*Helicobacter pylori* eradication with 7 days of levofloxacin-based first line therapy was safe and equal compared to 7 days of standard first-line therapy
Xiao [28]	2014	9	First-line	Levofloxacin	The 10-day levofloxacin-based triple therapy may be considered as an alternative for increasing cure rate of *H. pylori* infection in European areas. In Asian countries, standard triple regimen is still superior to levofloxacin-based therapy as first-line regimen for *H. pylori* eradication
Ye [29]	2014	10	First-line	Levofloxacin	Levofloxacin-based therapy was as safe and effective as triple therapy for *H. pylori* infection and could be considered as an additional treatment option
Li [30]	2015	8	First-line	Levofloxacin	Comparison of different eradication treatments for *H. pylori* showed that concomitant treatments, 10 or 14 days of probiotic supplemented triple treatment, 10 or 14 days of levofloxacin based triple treatment, 14 days of hybrid treatment, and 10 or 14 days of sequential treatment might be better alternatives for the eradication of *H. pylori*
Chen [31]	2016	41	First-line & rescue	Levofloxacin	The efficacy of levofloxacin triple therapy has been lower than 80% in many countries and it is not recommended when the levofloxacin resistance is higher than 5-10%
Zhang [32]	2017	17	Rescue	Levofloxacin	Comparing with bismuth-based quadruple therapy, levofloxacin-based triple therapy has higher eradication rate, compliance rate and lesser side effects, so we recommend it as a second-line rescue therapy
Yeo [33]	2019	27	Rescue	Quinolones	Quinolone-based bismuth-containing quadruple therapies for 10 days or more are the optimum second-line regimens for *H. pylori* eradication
Mori [34]	2020	3	Rescue	Sitafloxacin	Changes in the rate of antibiotic resistance to *H. pylori* were not observed from 2009 to 2015. The status of *gyrA* mutation is a superior marker for predicting successful eradication in STFX/AMX-containing triple regimen as a third-line rescue therapy

PPI: proton pump inhibitor.

**Table 2 molecules-25-05084-t002:** Studies evaluating the efficacy of a combination of a proton pump inhibitor, amoxicillin, levofloxacin and bismuth for the eradication of *H. pylori* infection.

Author	Year	Country	Treatment Order	Duration (Days)	Eradication n/N (Intention-To-Treat, %)
Bago [85]	2007	Croatia	First	7	57/66 (86%)
Cao [86]	2015	China	Frist	14	117/141 (83%)
Fu [87]	2017	China	First	14	167/200 (84%)
Gan (a) [77]	2018	China	First	14	155/200 (78%)
Gan (b) [77]	2018	China	First	14	155/187 (83%)
Gao [88]	2010	China	First	10	60/72 (83%)
Gisbert [89]	2015	Spain	Second	14	180/200 (90%)
Hsu [90]	2008	Taiwan	Third	10	31/37 (84%)
Aksoy [91]	2017	Turkey	First	14	93/111 (84%)
Liao [80]	2013	China	First	14	70/80 (88%)
Song [92]	2016	China	Second	14	97/132 (74%)
Yee [93]	2007	China	≥Second	7	37/51 (73%)

Gan (a): levofloxacin 500 mg/24 h; Gan (b): levofloxacin 200 mg/12 h.

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
