# Peer review of "Optimization Strategies Aimed to Increase the Efficacy of Helicobacter pylori Eradication Therapies with Quinolones"

_molecules, 2020, doi:10.3390/molecules25215084_

Round 1
Reviewer 1 Report
emergence of resistant bacteria and the patients with penicillin allergy, regimens other than the first line regimen are needed. In this manuscript, authors investigated the efficacy of quinolone including regimen. However, the manuscript is written well, some specific concerns are as follows.
- In ACG clinical guideline, both of bismuth quadruple therapy and levofloxacin regimen are recommended as the first line therapy ( J. Gastroenterol. 2017, 112, 212-239). In this manuscript, authors mentioned as a quinolone (mainly levofloxacin)-containing regimen is a safe and simple alternative to bismuth quadruple therapy in patients whose previous H. pylori eradication therapy has failed. If so, authors should show eradication rate, adverse effects, and tolerance rate of a quinolone -containing regimen and bismuth quadruple therapy.
- Authors mentioned as resistance to quinolones, in particular to levofloxacin is easily acquired, and in countries with a high consumption of these drugs, the resistance rate is increasing. When choosing regimen, resistance against antibiotics is important factor. Authors should discuss more about prevalence of quinolone resistance in the world with showing the resistance rate in each country.
- The main mechanism of quinolone resistance is mutation of gyrA gene in pylori. Authors should discuss more about this point with citing Helicobacter 2016 21(4):286-94 and United European Gastroenterol J. 20175(6):796-804.
Author Response
Dear Sir,
Enclosed you will find the revised paper in which the modifications suggested by the Reviewers have been made and are detailed as follows:
Reviewer 1:
In this manuscript, authors investigated the efficacy of quinolone including regimen. However, the manuscript is written well, some specific concerns are as follows.
1. In ACG clinical guideline, both of bismuth quadruple therapy and levofloxacin regimen are recommended as the first line therapy ( J. Gastroenterol. 2017, 112, 212-239). In this manuscript, authors mentioned as a quinolone (mainly levofloxacin)-containing regimen is a safe and simple alternative to bismuth quadruple therapy in patients whose previous H. pylori eradication therapy has failed. If so, authors should show eradication rate, adverse effects, and tolerance rate of a quinolone -containing regimen and bismuth quadruple therapy.
Thank you very much for your comment. In this respect, we would like to emphasize that:
In the section entitled “META-ANALYSES EVALUATING THE EFFICACY OF QUINOLONE-BASED REGIMENS FOR H. PYLORI ERADICATION” we point out that:
“Several meta-analyses have evaluated the efficacy of quinolone-based H. pylori eradication regimens, their main characteristics and conclusions being summarized in Table 1”. In this table, in several of the studies, it is stated that quinolone-based therapies are more effective and better tolerated than the bismuth quadruple therapy.
Furthermore, in the text, we state that “Initially, in 2006, two meta-analyses suggested that, after failure of treatment to eradicate H. pylori, a levofloxacin-containing rescue regimen was at least equally (and perhaps even more) effective, and better tolerated, than the generally recommended bismuth-containing quadruple regimen (i.e. a PPI, bismuth, tetracycline, and metronidazole)”.
On the other hand, in the section entitled “ADVANTAGES OF LEVOFLOXACIN-BASED REGIMENS FOR H. PYLORI ERADICATON” we include relevant information (regarding efficacy, tolerance and safety) of the quinolone-containing regimen vs. bismuth quadruple therapy: “As previously mentioned, administration of the classic bismuth-containing quadruple regimen is complex (tetracycline is usually prescribed every 6 hours and metronidazole every 8 hours). On the other hand, levofloxacin-based regimens (with the PPI and amoxicillin administered twice daily, and levofloxacin every 24 hours) represent an encouraging simpler alternative”. And also: “Finally, it should be stressed that, as previously mentioned, several meta-analyses have confirmed a lower incidence of adverse effects with levofloxacin-containing regimens compared with the classic bismuth quadruple combinations”.
2. Authors mentioned as resistance to quinolones, in particular to levofloxacin is easily acquired, and in countries with a high consumption of these drugs, the resistance rate is increasing. When choosing regimen, resistance against antibiotics is important factor. Authors should discuss more about prevalence of quinolone resistance in the world with showing the resistance rate in each country.
Thank you for your suggestion. Accordingly, we have now added the following information in the section “H. PYLORI RESISTANCE TO QUINOLONES”: “In a recent systematic review on the prevalence of resistance of H. pylori to antibiotics in different countries, the overall calculated levofloxacin resistance rate was 16%, although the figures varied significantly depending on the continent[40]. In particular, the prevalence rate was higher in Europe (24%) as compared to Asia (11.6%), and it was absent in Africa[40]. In Asia, different values among countries were detected, the resistance rate being 14.9% in Japan, 11.9% in Taiwan, and 2.6% in Hong Kong[40]. The prevalence of primary antibiotic resistance of H. pylori in 2008 and 2009 in 18 European countries was assessed in a multicenter study, and the rate for levofloxacin was 14%[3]. More recently, Savoldi et al performed a systematic review and meta-analysis to assess the distribution of H. pylori resistance to commonly used antibiotics[41]. In particular, primary levofloxacin resistance was ≥15% in all WHO regions, except in the European region (11%). Thus, this figure was 15% in the Americas region, 19% in the Eastern Mediterranean region, 30% in the Southeast Asia region, and 22% in the Western Pacific region[41]”.
3. The main mechanism of quinolone resistance is mutation of gyrA gene in pylori. Authors should discuss more about this point with citing Helicobacter 2016 21(4):286-94 and United European Gastroenterol J. 20175(6):796-804.
We agree with the suggestion. Accordingly, we have now included the two suggested references and we have added the following information in the section “H. PYLORI RESISTANCE TO QUINOLONES”: “The main mechanism of quinolone resistance is mutation of gyrA gene in H. pylori. Thus, gyrA mutation status has been demonstrated to be an important factor in predicting successful eradication with sitafloxacin-containing therapies[44, 45]. In this respect, a recent meta-analysis concluded that the status of gyrA mutation was a superior marker for predicting successful eradication in sitafloxacin/amoxicillin-containing triple regimen as a third-line rescue therapy[34]”.
We thank you for the comments and suggestions made, trusting that the manuscript is now suitable for publication.
Yours faithfully
Reviewer 2 Report
The review article by J. Gisbert is very interesting in terms of latest terapies against H. pylori, in particular when quinolone-based treatments are considered.
The author describes results on clinical studies using different alternatives and regimes after initial H. pylori eradication failures.
It is my opinion, however, that the article is not within the scope of a chemistry devote journal, as Molecules. Indeed, no relations with chemical, physicochemical properties of quinolone antibiotics are described, nor the relationship between any dynamics or structural characteristic in the mode of action of quinolones that explains the observed results after treatment. The review article molecules-980665 gives only data on clinical aspects of the quinolone-base therapy, without any conection with the topics included in the call.
Author Response
Dear Sir,
Enclosed you will find the revised paper in which the modifications suggested by the Reviewers have been made and are detailed as follows:
Reviewer 2:
The review article by J. Gisbert is very interesting in terms of latest terapies against H. pylori, in particular when quinolone-based treatments are considered.
The author describes results on clinical studies using different alternatives and regimes after initial H. pylori eradication failures.
It is my opinion, however, that the article is not within the scope of a chemistry devote journal, as Molecules. Indeed, no relations with chemical, physicochemical properties of quinolone antibiotics are described, nor the relationship between any dynamics or structural characteristic in the mode of action of quinolones that explains the observed results after treatment. The review article molecules-980665 gives only data on clinical aspects of the quinolone-base therapy, without any conection with the topics included in the call.
Thank you for the comment. As it is stated in the Introduction section, the aim of the present review is to summarize the role of quinolones in the management of H. pylori infection, mainly focusing on the optimization strategies aimed to increase the efficacy of quinolone-based eradication therapies. In fact, we agreed the content of our review with the Editors before submission, emphasizing its clinical perspective. Nevertheless, following the Reviewer’s suggestion, we have now included a completely new section entitled “PHARMACOKINETICS OF QUINOLONES”, where we explain the quinolones behavior on the basis of their pharmacokinetic properties.
We thank you for the comments and suggestions made, trusting that the manuscript is now suitable for publication.
Yours faithfully
Reviewer 3 Report
This is a review by one of the world’s authorities on H pylori therapy. Unfortunately, it does not conform to the principles of antimicrobial stewardship and largely uses data derived from undefined populations that achieved less than optimal results even for the better regimen. They also use the term optimize or optimization in a literal way rather than as it is used in relation to therapy of an infectious disease or as used according to the principles of antimicrobial stewardship. Optimization in modern infectious disease therapy is related to outcome which is defined in terms of cure rates especially in relation to 100%. This manuscript is written based on improvements irrespective of whether the actual outcome is near what is achievable. In 2011 it was shown that increasing the duration of fluoroquinolone triple therapy to 14 days would reliably achieve cure rates of 95% or greater with susceptible infections. Those results were subsequently confirmed ending the search of optimum duration. This issue was reopened as if it were unsettled. The dosing and effect of potency of the PPI in fluoroquinolone triple therapy remained to be optimized but are more minor issues as they can be dealt with existing data. Since fluoroquinolone resistance is all or none, there is no place to optimize in the presence of resistant infections. Current guidelines (e.g., Houston and Taipei consensus statements) state that fluoroquinolone, metronidazole, and clarithromycin-containing triple therapies should not be used empirically with the caveat that they could be considered if resistance was proven to be rare in a region.
The author has not dealt with the issue of FDA warning about use of fluoroquinolones and the multiple Black Boxes that they have accumulated. As such a recent study in China that used a susceptibility-based algorithm to assign therapy put fluoroquinolone triple therapy last. The lack of a discussion of this very serious issue must be dealt with to prevent the serious consequences noted by the FDA.
Susceptibility testing can now be reliably be done on stools. One does not need to test every patient rather one needs to know the local and regional rates of resistance. If high, (e.g., >5% or 10%), then the individual susceptibility testing would be recommended. Until local or regional susceptibility is known, one can use test of cure data to estimate the local resistance. Failure to reliably achieve cure rates of >90%, preferably higher would indicate that the empiric use of the combination was unwise. Most of the data shown in the manuscript come from regions where resistance levels were likely high than allowable or the regimens were suboptimal and should not have been used.
It is clear that the addition of bismuth improves the outcome of some H pylori therapies but whether the outcome is additive or synergistic is unclear with most evidence coming down on the side of additive. Since it is possible to reliably achieve very high cure rates with 14-day fluoroquinolone triple therapy, the role for bismuth is unclear.
Author Response
Dear Sir,
Enclosed you will find the revised paper in which the modifications suggested by the Reviewers have been made and are detailed as follows:
Reviewer 3:
This is a review by one of the world’s authorities on H pylori therapy. Unfortunately, it does not conform to the principles of antimicrobial stewardship and largely uses data derived from undefined populations that achieved less than optimal results even for the better regimen. They also use the term optimize or optimization in a literal way rather than as it is used in relation to therapy of an infectious disease or as used according to the principles of antimicrobial stewardship. Optimization in modern infectious disease therapy is related to outcome which is defined in terms of cure rates especially in relation to 100%. This manuscript is written based on improvements irrespective of whether the actual outcome is near what is achievable. In 2011 it was shown that increasing the duration of fluoroquinolone triple therapy to 14 days would reliably achieve cure rates of 95% or greater with susceptible infections. Those results were subsequently confirmed ending the search of optimum duration. This issue was reopened as if it were unsettled. The dosing and effect of potency of the PPI in fluoroquinolone triple therapy remained to be optimized but are more minor issues as they can be dealt with existing data. Since fluoroquinolone resistance is all or none, there is no place to optimize in the presence of resistant infections. Current guidelines (e.g., Houston and Taipei consensus statements) state that fluoroquinolone, metronidazole, and clarithromycin-containing triple therapies should not be used empirically with the caveat that they could be considered if resistance was proven to be rare in a region.
We agree with the Reviewer’s comments, and we have now added his/her suggestion in the text (please see H. PYLORI RESISTANCE TO QUINOLONES section): “Furthermore, some consensus (such as Houston and Taipei consensus statements) state that fluoroquinolone, metronidazole, and clarithromycin-containing triple therapies should not be used empirically with the caveat that they could be considered if resistance was proven to be rare in a region”. We have also included the two suggested references (Houston and Taipei consensus statements) in the reference list.
The author has not dealt with the issue of FDA warning about use of fluoroquinolones and the multiple Black Boxes that they have accumulated. As such a recent study in China that used a susceptibility-based algorithm to assign therapy put fluoroquinolone triple therapy last. The lack of a discussion of this very serious issue must be dealt with to prevent the serious consequences noted by the FDA.
We agree with the suggestion, and we have now added detailed information on this topic (please see section entitled ADVANTAGES OF LEVOFLOXACIN-BASED REGIMENS FOR H. PYLORI ERADICATON): “In 2018, the United States Food and Drug Administration (FDA) and the European Medicines Agency (EMA) issued new warnings about serious adverse effects of fluoroquinolones. Both the FDA and the EMA demanded a change in the labelling of the entire class of antibiotics highlighting these new risks. They also discouraged the use of fluoroquinolones for most mild and moderate infections or where there is a therapeutic alternative, restricting their use exclusively to serious infections such as pneumonia, anthrax and plague, or other non-self-limiting infections in which the therapeutic benefit outweighs the risks. Nevertheless, it is important to remember that H. pylori causes a chronic infection that can trigger serious conditions such as peptic ulcer (possibly with complications, such as gastrointestinal bleeding) and gastric cancer, and that treatment requires the combined use of several antibiotics. In any case, this health authority recommendation supports the responsible use of antibiotics and the need to carefully analyze the available therapeutic options and their risk/benefit ratio before prescribing any drug”.
Susceptibility testing can now be reliably be done on stools. One does not need to test every patient rather one needs to know the local and regional rates of resistance. If high, (e.g., >5% or 10%), then the individual susceptibility testing would be recommended. Until local or regional susceptibility is known, one can use test of cure data to estimate the local resistance. Failure to reliably achieve cure rates of >90%, preferably higher would indicate that the empiric use of the combination was unwise. Most of the data shown in the manuscript come from regions where resistance levels were likely high than allowable or the regimens were suboptimal and should not have been used.
We agree. Accordingly, we have now added the following sentence to clarify this issue (please see HOW TO OPTIMIZE QUINOLONE-BASED TREATMENTS FOR H. PYLORI ERADICATION? section): “Nevertheless, it is not needed to test every patient but rather it would only be necessary to know the local and regional rates of resistance (if high, then the individual susceptibility testing would be recommended)”.
It is clear that the addition of bismuth improves the outcome of some H pylori therapies but whether the outcome is additive or synergistic is unclear with most evidence coming down on the side of additive. Since it is possible to reliably achieve very high cure rates with 14-day fluoroquinolone triple therapy, the role for bismuth is unclear.
We agree. Accordingly, we have now added the following sentence to clarify this issue (please see HOW TO OPTIMIZE QUINOLONE-BASED TREATMENTS FOR H. PYLORI ERADICATION? section, addition of bismuth subsection): “Nevertheless, whether the outcome of adding bismuth is really additive or merely synergistic is still debated, and therefore the role of adding bismuth to the conventional 14-day levofloxacin-based triple regimen remains unclear.
We thank you for the comments and suggestions made, trusting that the manuscript is now suitable for publication.
Yours faithfully
Round 2
Reviewer 1 Report
Well revised manuscript. No further comments.
Reviewer 2 Report
In the new version the author included a section devoted to the pharmacokinetic properties of quinolones, as a contribution to the topic related with biological chemistry.
In my opinion the review article is quite interesting for summarizing and evaluating new information on clinical therapies based on quinolone antibiotics, but little connection with chemistry can be found, even taken biological chemistry or pharmaco-chemistry, in the broader sense.
However, as the author commented, the clinical topics covered in the manuscript, i.e. the defined profile of this contribution was a desired effect already settled with the editor of this special issue, so, I think the whole history available for this submission gives enough elements for the final editor decision.
Reviewer 3 Report
The fluoroquinolone paper is greatly improved. However, it is still focused on past when trial and error and good enough were the standards rather than embracing Antimicrobial Stewardship where outcome is king.
Optimization in this paper is picking the best from a set of poor results. Those meta-analyses show that the concept of comparing poorly performing regimens or even excellent regimens in populations with resistant germs both yield poor results. One may be better than another but using a universal standard such as must achieve 95% or even 90% cured to be acceptable, they all fail. Optimization under antimicrobial stewardship means being able to reliably achieve high cure rates (typically 95% or greater). One could further qualify this to in adherent patients or mITT such as those what had assessment of outcome but the focus is the same. Which was how well we did, not how to do well. For every trial the actual results must be given (a was better than b) could be 67% vs. 72% or 93% vs. 97%. In one both failed in the other both were good but one was better. For example, regarding ref 72 we are told that 14 day therapy was significantly better than 7 day therapy: in itself that is meaningless. We were not told that that they achieve 95% cure rates. Ever statement regarding better or worse must provide the cure rates.
Fluoroquinolones can be used as primary or after failure with another therapy. The fact that metronidazole or clarithromycin resistance do not affect fluoroquinolone therapy is irrelevant as there is no cross resistance. Fluoroquinolones have been shown to be highly effective in population were resistance to antibiotics in general is very high if they were given in a susceptibility-based program using known highly effective regimens (e.g., 14 day). Every place in the world has local susceptibility data available. If they do test of cure then they know that either resistance is high or patients are non-compliant (or both). Thus susceptibility can be assessed prospectively or retrospectively. The author suggest they throw up their hands and try something. This is infectious disease therapy where with optimized therapies (i.e., proven to reliably produce high cure rates not be a little better than a really poor regimen) failure to achieve high cure rates is immediately known (resistance or compliance issues).
We are told 10-14 day therapy. Both cannot be recommended unless they are equivalent and then one would use 10 days. Optimum does not allow a range. Do both reliably achieve high cure rates (e.g., greater than a defined number over 90%?). Has that been proven in a head-to-head non-inferiority trial? This is how infectious disease works.
The recommendations should include obtaining test of cure and using that data to define, proof, and survey what works locally. When it begins to fail, change. The (Chen, Long et al. 2019) paper clearly showed how to use levofloxacin as a first line therapy. The logic in choosing the order to use triple therapies is clearly spelled out and incorporates the FDA and European drug use cautions.
The drugs are used with caution in the text but are safe in the abstract
Chen, Q., X. Long, Y. Ji, X. Liang, D. Li, H. Gao, B. Xu, M. Liu, Y. Chen, Y. Sun, Y. Zhao, G. Xu, Y. Song, L. Yu, W. Zhang, W. Liu, D. Y. Graham and H. Lu (2019). "Randomised controlled trial: susceptibility-guided therapy versus empiric bismuth quadruple therapy for first-line Helicobacter pylori treatment." Aliment Pharmacol Ther 49(11): 1385-1394.